# Changes in Land Relief in Urbanised Areas Using Laser Scanning and Archival Data on the Example of Łódź (Poland)

**Marcin Jaskulski** [1,*], **Iwona Jażdżewska** [1] and **Aleksander Szmidt** [2]

1 Institute of Urban Geography, Tourism and Geoinformation, Faculty of Geographical Sciences, University of Lodz, 90-139 Łódź, Poland; iwona.jazdzewska@geo.uni.lodz.pl
2 Department of Physical Geography, Faculty of Geographical Sciences, University of Lodz, 90-139 Łódź, Poland; aleksander.szmidt@geo.uni.lodz.pl
* Correspondence: marcin.jaskulski@geo.uni.lodz.pl

**Abstract:** The authors undertook to examine and analyse the changes in the relief in Łódź city centre over a period of over one hundred years. Archival cartographic resources containing morphometric information and contemporary laser scanning data (LIDAR) are used to analyse changes. This required appropriate transformation of these data to generate a differential relief map. Information on the geographical environment (waters, relief) is linked to the spatial development of the city. The analyses revealed several characteristic types of changes occurring in the area, which are presented in the form of case studies.

**Keywords:** laser scanning (LIDAR); geographic information systems; numerical terrain models; historical GIS; land relief changes; anthropopressure

## 1. Introduction

Relief is dynamic in nature, meaning that it is constantly changing under the influence of various external factors. It remains in the mutual interaction between the other elements of the natural environment and between humans and their activities. Depending on the processes shaping a given area, changes in this natural element may be very slow, noticeable only in studies covering a wide range of time, or very sudden, even posing a danger to humans and their activities. The problem of analysing and visualising changes in terrain relief was rarely presented in scientific and cartographic studies before the advent of Geographic Information Systems (GIS). The development of GIS techniques, particularly surface modelling and analysis, combined with the availability of newer and more accurate survey data such as laser scanning, has made this topic increasingly popular with geomorphologists and Quaternary geologists. There are many studies in the literature on relief changes that we can describe as innovative. Typical sites where such studies have been conducted are mining areas [1–3], seismic activity [4,5], glacier activity [6] anthropogenic disruption of fluvial processes through, for example, the construction of dams and retention reservoirs [7,8], or changes associated with the centuries-long and often turbulent history of large urban centres [9–17]). These studies often lack a comprehensive approach. It includes the construction of numerical terrain models and their adaptation to the state so that they can be compared, as well as the detection and comprehensive analysis of such changes based on historical sources.

The aim of this paper is to examine the changes in relief in the city centre area over a period of more than one hundred years. The Polish post-industrial city of Łódź was chosen for the study (Figure 1). To achieve the objective, it was necessary to obtain archival and contemporary data and develop a methodology to compare them. To this end, available archival cartographic material was searched for relief data. Unfortunately, the plans of the city from the first century of its industrialisation did not include hypsometric information. Therefore, the map prepared by Jasiński in 1917 was chosen as the basis for the study

of relief changes. It contains the area of the oldest part of the city. The contemporary relief was extracted from LIDAR data from 2017. The result of the research was to identify characteristic places where changes in relief occurred and to explain their causes. The main problem that the authors had to face was bringing the terrain models created from various data to a state that would allow their comparison. Then, the detection of changes in the relief of the area coming from and finding them in the topography and history of the city.

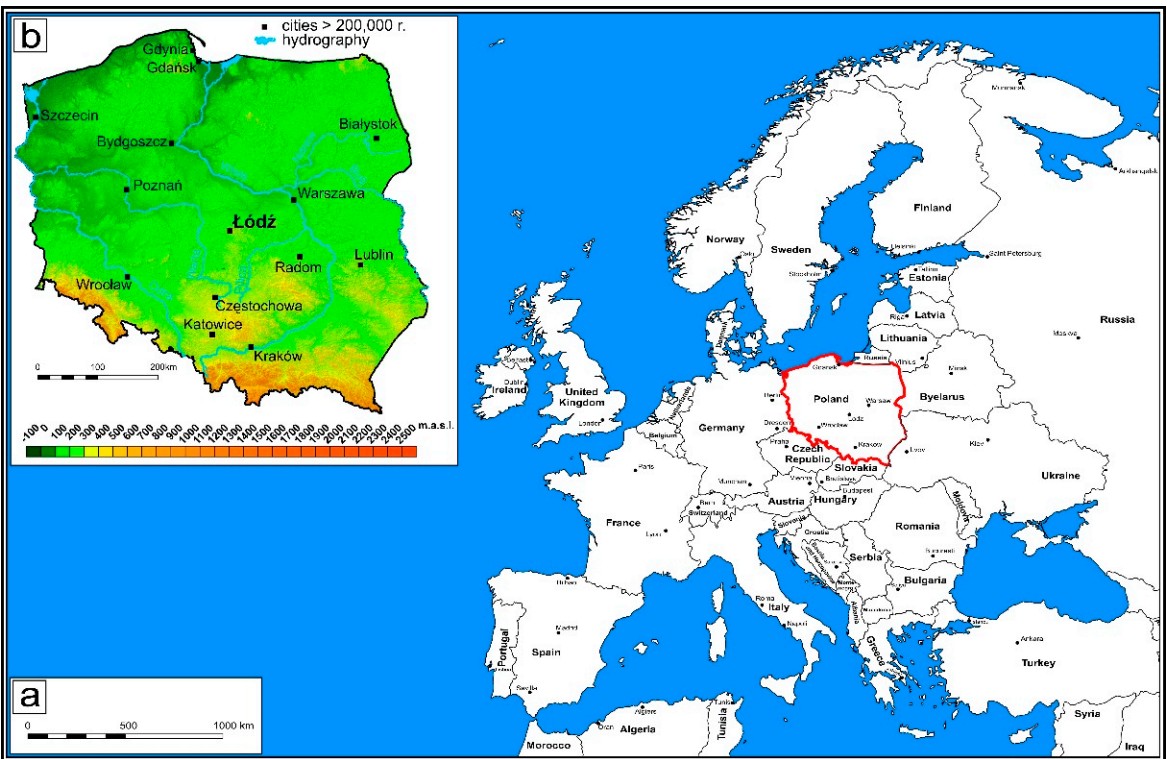

**Figure 1.** Location of the study area in Europe (**a**) and Poland (**b**). Source: Own elaboration based on GUGiK and Golden Software data.

## 2. Study Area

The study of relief changes was conducted in the centre of Łódź, the third-largest city in Poland with a population of almost 700,000 (2020) located in central Poland (Figure 1). It is one of the most unusual Polish cities, as its greatest development took place in the early 19th century, when the town was located in the Kingdom of Poland (under Russian jurisdiction), at a distance of 100 km from Prussia. It was then decided to locate clothmakers' settlements in small Polish towns and to give them opportunities for economic development. One of them was Łódź, which in 1829 had 4271 inhabitants living in 369 houses, mostly wooden [18]. The city was planned in the so-called "raw root" (Latin in cruda radice) to the south of a medieval agricultural town. In 1821–1823, the first clothmakers' settlement called "New Town" was established along the Łódka River, followed by another cotton and linen settlement called "Łódka" along the Jasień River (1824–1828). The city's location was attractive to German textile manufacturers, who wanted to enter the Russian market without having to pay import duties [19]. The centre of the town was divided by a grid of streets with several-story houses in the vicinity of which the owners located their craftsmen's workshops or factories. A characteristic feature of Łódź was the mix of industrial and residential functions. This applied to both rich industrialists and small craftsmen.

The city was attractive to clothmakers and cotton weavers, mainly from the German-speaking countries of the time [20]. Within 10 years, more than 1000 craftsmen families migrated to Łódź, settled in the city and started its spectacular industrial development. The

population continued to grow to almost 50,000 in 1875. It exceeded 300,000 inhabitants in 1902, 500,000 in 1921 and 600,000 in 1932 [21]. In 1913, Łódź was inhabited by approximately 460,000 people, and there were 35 large textile factories that employed over 500 people each, among them 17 large industrial plants employing over 1000 workers [22]. The owners of the largest factories were the following (by number of workers employed): K. Scheibler, I.K. Poznański, L. Geyer, J. Heinzel, J. Kunitzer, L. Grohmann, Sz. Rosenblatt (all in cotton production) and J. Heinzel, M. Silberstein, Allart, Rousseau and Co., K. Bennisch, F.W. Szweikert, Leonhard Woelker and Girbart, I. Richter, M. Kon, J. Wojdysławski (in wool production) [23].

It is worth mentioning that the religious structure of Łódź until 1939 was not homogeneous. The inhabitants represented mainly Catholic, Protestant and Jewish faiths. Depending on the period of the town's development, different religious groups had different shares in the population structure. In the middle of the 19th century, for example, Jews made up about 10% of the entire population, at the turn of the 19th and 20th centuries, 25%, and in 1914, they were over 30%. In the mid-19th century, Protestants and Catholics each amounted to 43.5%, after which the Protestant share began to decline and reached 13.5% in 1914 [21].

The wealth of the industrialists of Łódź may be evidenced by their successful efforts to bring in 1865 a branch line (Koluszki–Łódź Fabryczna railway station) with the first railway line on Polish soil—the Warsaw-Vienna Railway. The city gained connections to the economic centres of the Polish Kingdom, as well as to Russian markets. In addition, tracks were laid from the main line to K. Scheibler's and L. Grohman's factories. A few years later, in 1898, a decision was made to build a railway line that was to connect Warsaw with Łódź and Kalisz and run all the way to the Prussian border. The new line and Łódź-Kaliska station were opened in 1902 [24]. The 1st and 2nd World War fronts passed through the area of Łódź in the 20th century. However, the city itself and its buildings were not destroyed. Authors looking for traces of the former settlement faced such problems [25,26]. Łódź survived, and the work had to be carried out differently.

In terms of physical-geographical regionalisation [27,28], Łódź lies on the border of two mesoregions (Figure 2), i.e., the Łask Plateau (318.19) and the Łódź Hills (318.82). The study area is entirely covered by quaternary sediments, mainly clays and sands, of high thickness, reaching 120 m in places [29]. The culminations are made up of sands and gravels and glacial till from terminal moraines, often in the form of squeezes and accumulations. They are surrounded by areas made up of hydro-glacial sands [30–32]. The area is cut by a series of valleys and ravines filled with peat and silt. The central part of the city is much less morphometrically diversified and is built mainly of glacial till and hydro-glacial sands. The whole area is dotted with individual hills made up of gravel and sand from dead ice moraines. Towards the south-east, the proportion of sediments associated with river valleys increases. The highest areas are in the north-east of the city (278 m), while the lowest area is in the south-west (160 m). However, it should be borne in mind that in reality, in an urbanised area, a large part of the landforms has been levelled and the surface will be dominated mainly by anthropogenic sediments [33].

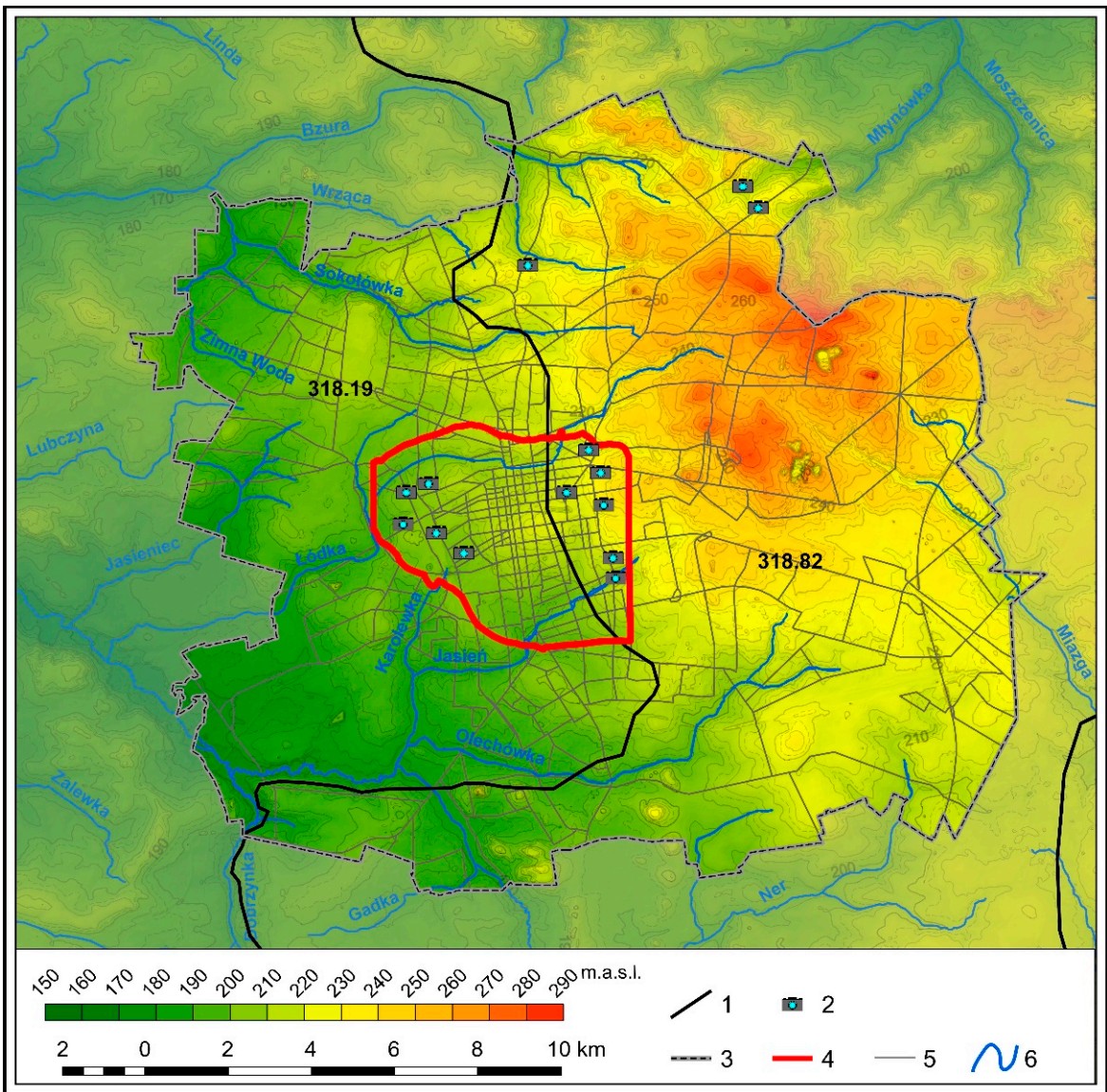

**Figure 2.** The relief of Łódź and its immediate surroundings. 1—boundaries of physico-geographical mesoregions; 2—location of photographs presented in the article; 3—administrative boundaries of Łódź; 4—scope of Jasiński's map (1917); 5—main roads; 6—watercourses. Source: Own study based on GUGiK and MPHP data.

Łódź lies on a watershed of the first order, from which many small rivers flow radially. It is estimated that the length of these watercourses within modern Łódź is about 77 km. It is supposed that in the pre-industrial period, this length may have been up to three times longer [32,34]. There were so many that a single stream could run by each textile factory [35]. Translating this into the language of geomorphology and geology, we can imagine an area with expansive hills (Figure 3a), dissected by numerous ravines (Figure 3b), where several spring niches were present (Figure 3c), where many watercourses started their course, which due to the steeper slope of the terrain, strongly eroded the surfaces (Figure 3d).

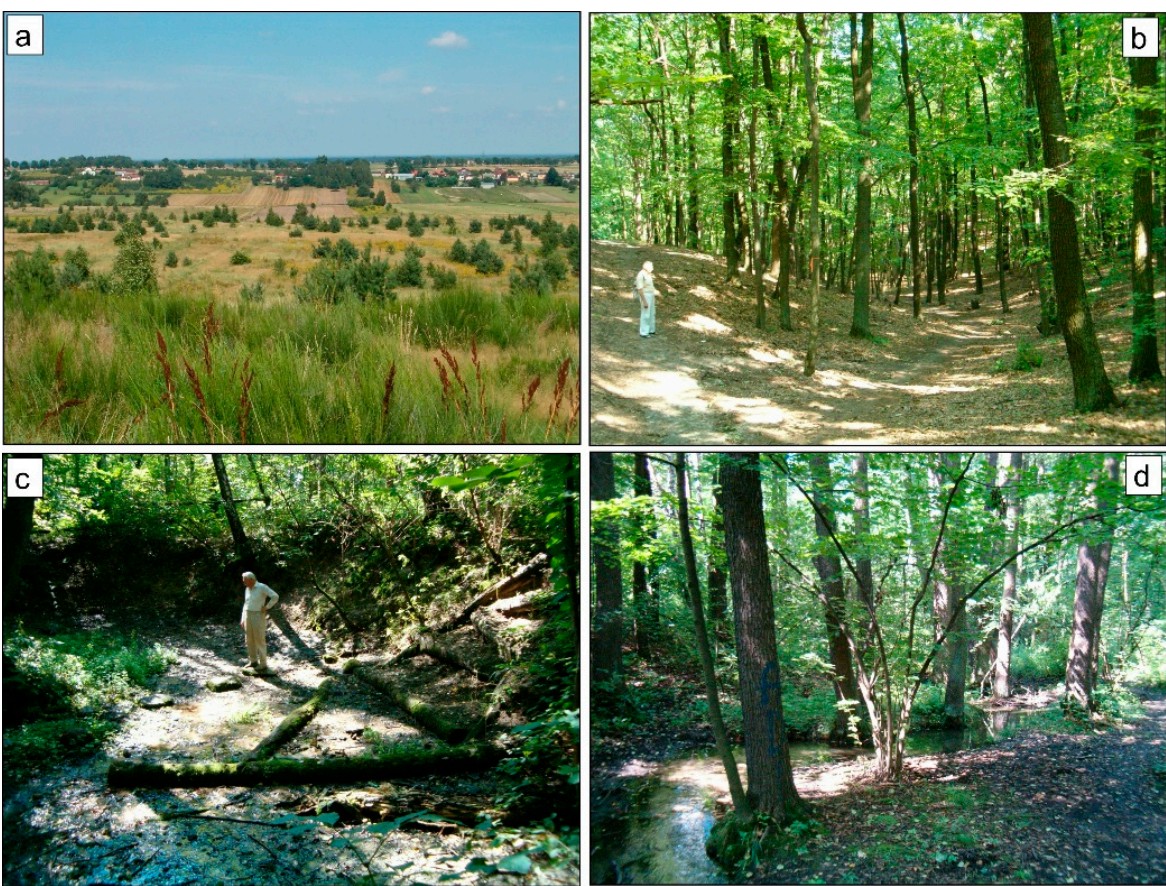

**Figure 3.** Examples of non-urbanised landscapes in the Łódź area. (**a**) edge zone of the Łódź Heights; (**b**) a ravine in Łagiewnicki Forest in the edge zone of the Łódź Heights; (**c**) lateral spring niche in the initial course of the Bzura River; (**d**) bottom of the Bzura River valley in its spring section (photo A. Szmidt 2006).

Away from the city centre, elements of its former landscape have been preserved, e.g., sections of valleys in their natural form covered with forest (Figure 3). They allow us to imagine what the landscape and relief of 19th century Łódź looked like. That is, rapid streams in deeply incised spring sections, waterlogged valley bottoms and extensive forested areas cut through by ravines. Depending on the location, there were hills of end moraines or dammed moraines and extensive surfaces and slopes made up of glaciofluvial sand and gravel. On the raw material and engineering geology side, the geological structure alone can give us an indication of possible future man-made surface transformations. The hills themselves, as convex forms, presented difficulties for the construction of the city's infrastructure, as they required the levelling or softening of their relief. As they were made up of sand and gravel, these formations were also sources of aggregate necessary for the construction of houses and factories or the covering of wetlands. In areas made of boulder clay, if the structure of the clay was right, brickyards could be built, the remnants of which could be the so-called clay pits.

## 3. Materials and Methods

The research procedure carried out in this paper included several stages (Figure 4). The first step was to develop the most optimal numerical terrain model describing the former relief of the city. It was further necessary to adapt modern morphometric data (LIDAR) for comparative analyses with archival data. Spatial Analyst tools included in ESRI ArcGIS 10.4 software were used to identify and compare differences between the

1917 and 2017 landforms. The results were interpreted based on archival plans of the city, photographs and aerial photographs and orthophotos.

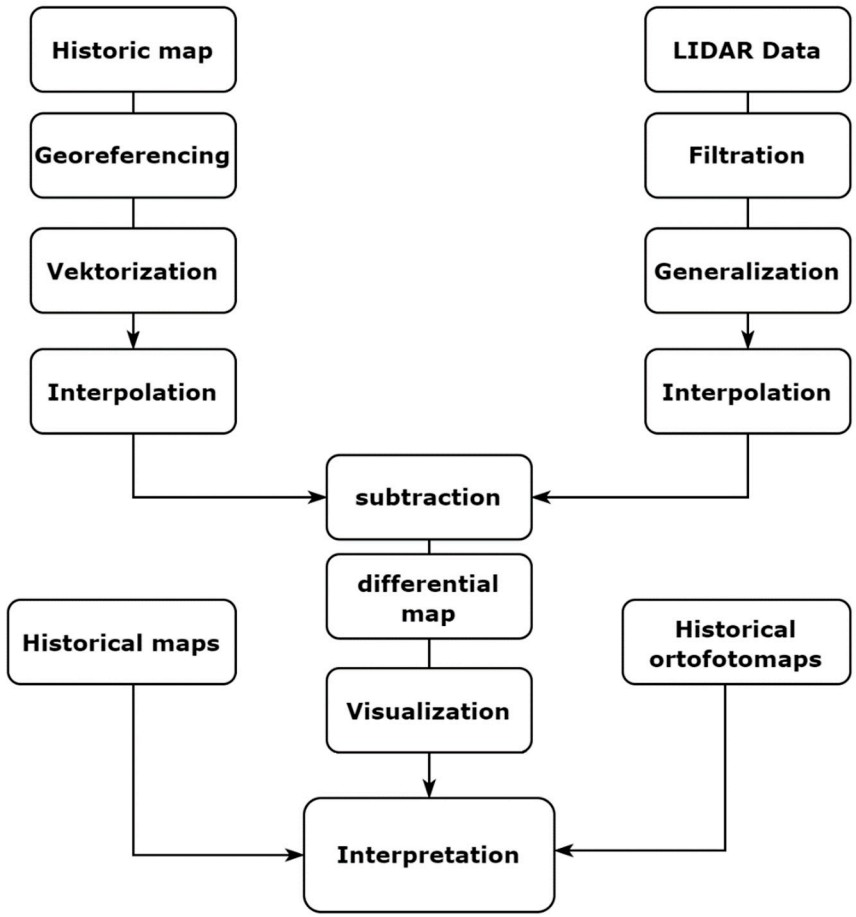

**Figure 4.** Research procedure diagram.

### 3.1. Historical Model

To find the oldest maps and plans of the city, a search was made for available archival cartographic studies. The authors had an easy task, as in 2020 most of them were made available in the form of the Historical Atlas of the City of Łódź for Science, Education, Culture, Economy and Society (2019) and a related application compiling maps and aerial photographs from different years. The search carried out showed that in the case of Łódź a small number of historical maps containing information on the relief of the land surface are available. Significantly, there is a lack of studies of this type from the 19th century and beyond.

The oldest source of data showing the relief of Łódź is a topographic map at a scale of 1:200,000 (Topograph. Special—Karte von Mittel—Europa. This map dates to the late 19th century (1883–1909). It contains information on the land surface relief described using the Lehman method (1750–1815) [36]. This method can be used to represent height ranges of slopes, slope exposures, individual height points. Due to the scale and quality of the data, a map of this type was not usable.

Other sources of data were the plans of the city drawn up by Filip de Viebig in 1823, Majewski (1889), Starzyński (1897), Lindley (1909), Romer (1923). They did not contain information about the terrain but presented the various stages of the town's layout and development. The best source of morphometric information was maps elaborated by Jasiński in 1917 at a scale of 1:4000 (Figure 5). These maps were made based on accurate surveying techniques used for the design of the water and sewage systems of

Łódź developed by W. Lindley in 1902. The relief of the land surface is described in 1-m contour cuts. The limited spatial coverage of the study, defined by the purpose of the city's sewerage network at the time and the city limits, can be considered a disadvantage of this data source.

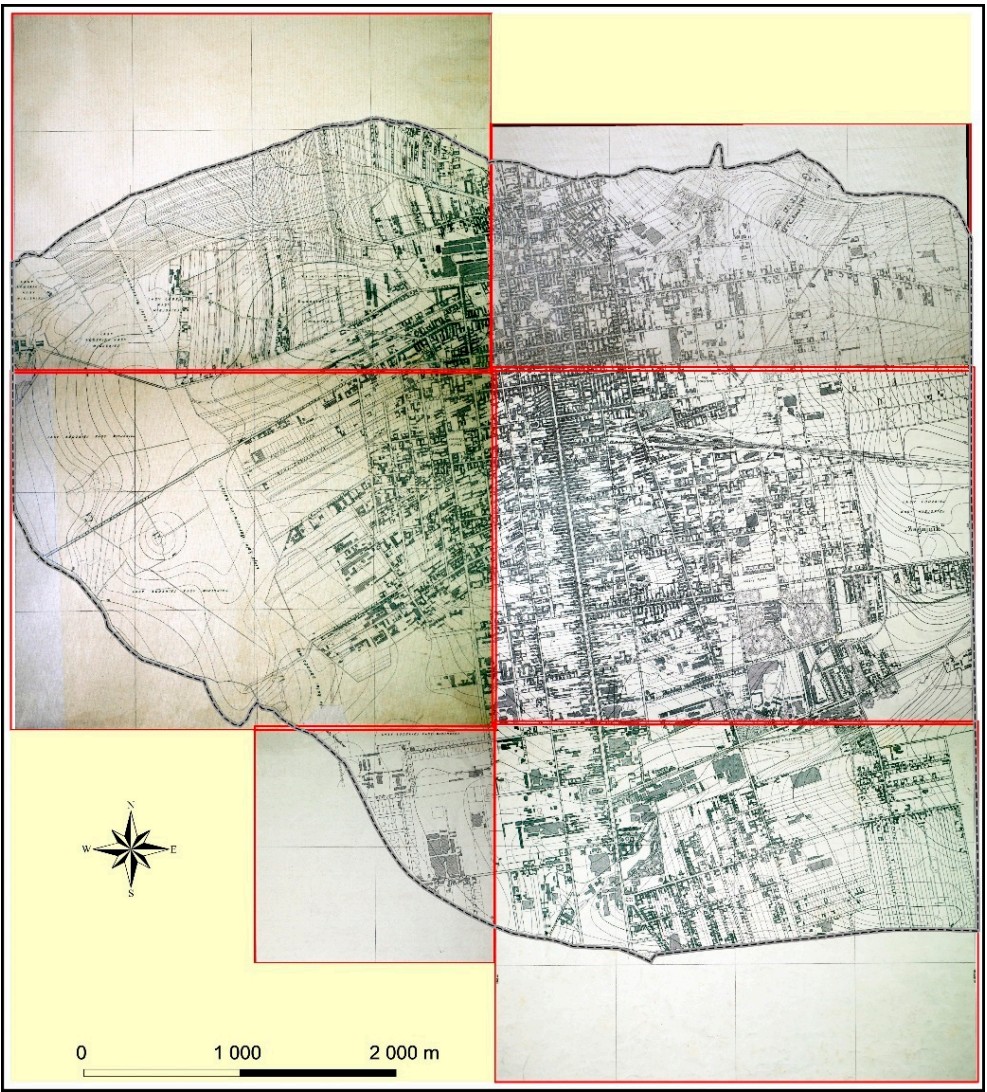

**Figure 5.** Compilation of Jasiński's 1917 map sheets. Source: State Archive in Łódź.

Due to its accuracy, this map was chosen by the authors as the basis for creating a numerical terrain model for the former Łódź. The map was obtained in the form of scans of six sheets from the State Archive in Łódź. (Figure 5). Spatial reference was given by linking landmarks on Jasiński's map to those on contemporary maps and aerial photographs that already had georeferencing. It was easiest to identify such places in the city centre, where most of the buildings have survived to this day. Problems occurred with the locations of these points on the former periphery of the city [37], which have undergone the greatest spatial transformation over the last few decades. The next stage of the work was the interpretation and screen digitisation of the contour drawing.

After the digitisation process, the correct marking and sequencing of the values of the individual contours were verified again. This procedure was followed by the creation of a Numerical Terrain Model. It should be noted that the contour drawing, by its nature, has an uneven horizontal distribution of contour lines and requires different processing to NMT than data from automatic surveys. Based on trials, the optimum grid resolution of 10 m was determined (minimum number of artefacts after the interpolation process and

faithful secondary representation of the contour lines in relation to the original drawing on Jasiński's map (1917)). In case of inconsistencies, the level drawing was enriched with additional morphometric information based on geomorphological analysis. The main task was the addition of interpreted selected skeletal lines of the terrain, which are used to define the parameters of concave or convex forms [38]. Ordinary kriging with spherical semivariogram was chosen for the interpolation process. The NMT for 1917 generated this way is shown in Figure 6.

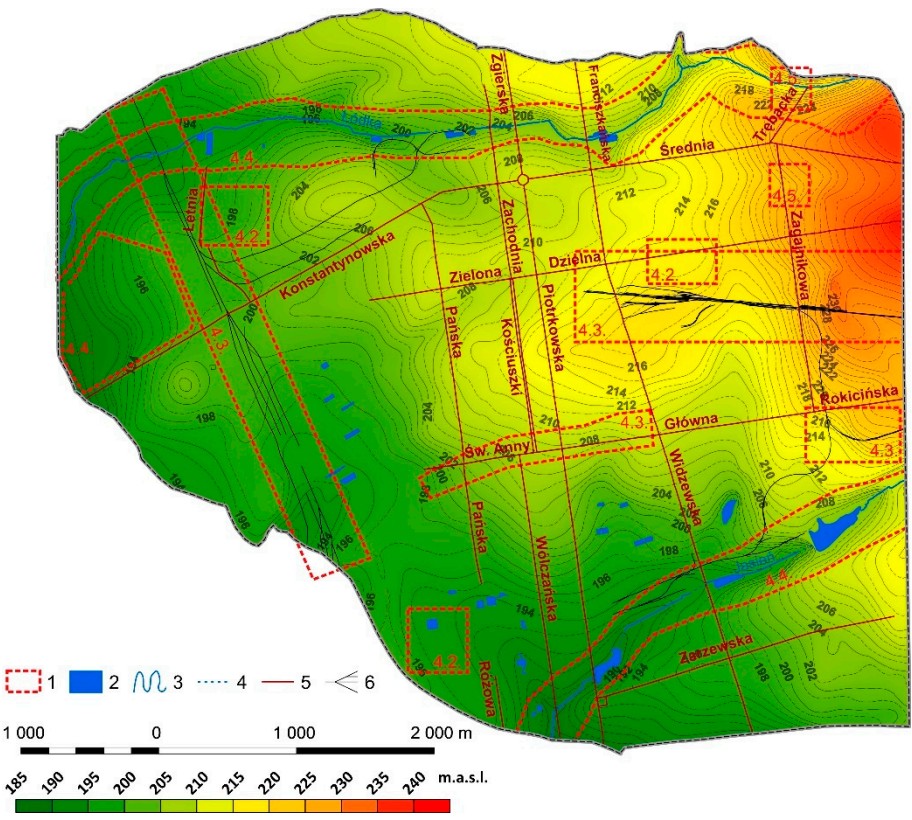

**Figure 6.** The relief of a fragment of Łódź based on Jasiński's map (1917). 1—areas of detailed analysis, 2—waters, 3—surface watercourses, 4—underground watercourses, 5—main streets, 6—railway tracks (situation in 1917). Source: Own study.

Then, 85 points were selected with precisely defined elevation and location (geodetic reference points), and on this basis, the vertical RMS (Root Mean Square) error was calculated using commonly available methodology and formulae [39]. The error value was 0.43 m. This result is incomparably better in the analyzed case than that obtained using SAR (synthetic aperture radar) interferometry, which was 2.3 m [40]. This proves the high accuracy of the data collected by traditional geodetic methods in the field, on which the historical model was built.

### 3.2. The Contemporary Model

Several global and local numerical terrain and land cover models are available for the Łódź area, made using various measurement techniques and resolutions, such as SRTM, GDEM, or NMT25 with a horizontal resolution of at least 25 m. For the analysis and presentation of the contemporary relief of the city, morphometric data included in the IT System of National Protection (ISOK), derived from laser scanning measurements (LIDAR), were selected. A filtered model is made available by the General Office of Geodesy and Cartography (GUGiK) with a horizontal resolution of at least 1 m and vertical accuracy of 20 cm.

The chosen data model had the highest data quality, but it also had shortcomings. The first problem relates to data for built-up areas. When processing the source data, NMT providers usually reinterpolated the data based on measurements from surrounding areas after removing the content related to the presence of buildings. The finished elevation model shows clear geometric traces at the locations of the buildings, distorting the actual relief image. Therefore, when differential models are created with highly smoothed surfaces (e.g., data from a contour drawing), differences in the shape of the surface in the form of polygons may arise.

The second problem was related to the greatest advantage of the LIDAR model, namely, its high horizontal and vertical resolution. When comparing high-quality data with the data from the smoothed historical model, differences appeared in places where the modern relief was mapped more faithfully than in the historical model. This could only be due to differences in the resolutions of the processed rasters.

To solve these problems, it was decided to generalise the contemporary model to the resolution of the model depicting archival data while maintaining the division according to rectangular coordinates of the basic field (Figure 7). This ensured that the averaged values in each pixel were perfectly within the geometric parameters above each other.

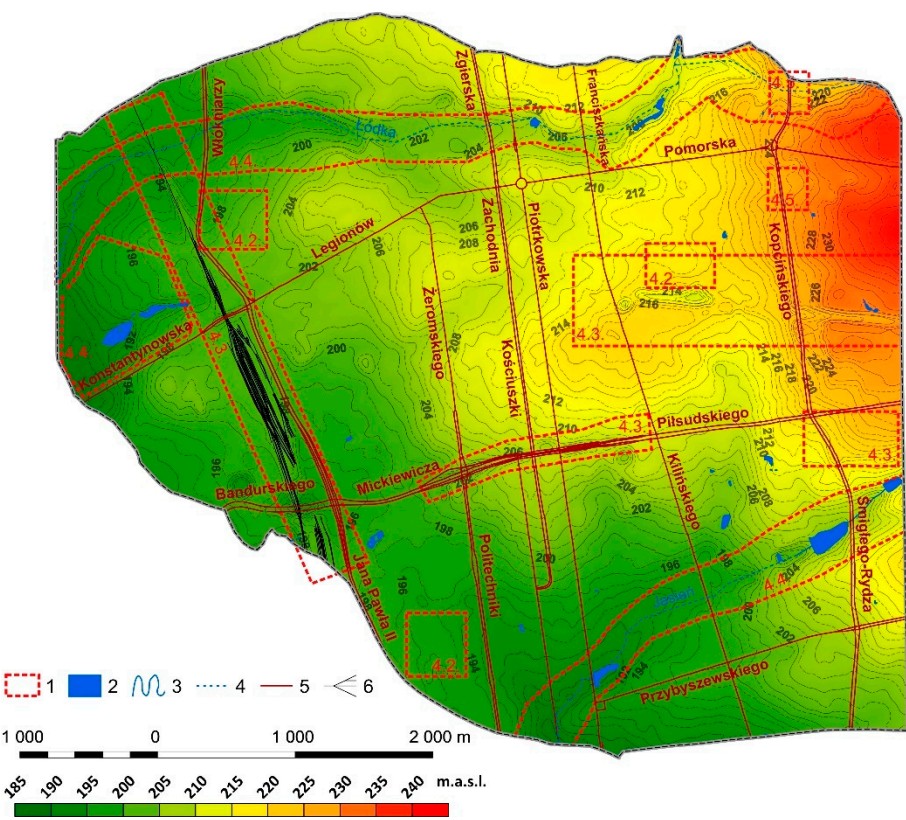

**Figure 7.** The relief of the modern land surface of a fragment of Łódź based on generalised laser scanning data from 2017. 1—areas of detailed analysis, 2—waters, 3—surface watercourses, 4—underground watercourses, 5—main streets, 6—railway tracks (situation in 2017). Source: Own study.

As in the case of constructing the model from the historical period, the RMS error was calculated on the basis of 85 control points, which in this model was 0.34 m. Due to the similar accuracy, this result allowed for comparisons.

A differential map was generated from the prepared two models (Figure 8), showing changes in the relief of the study area over a period of 100 years (ArcGIS 10.4 and the Spatial Analyst Tools function were used).

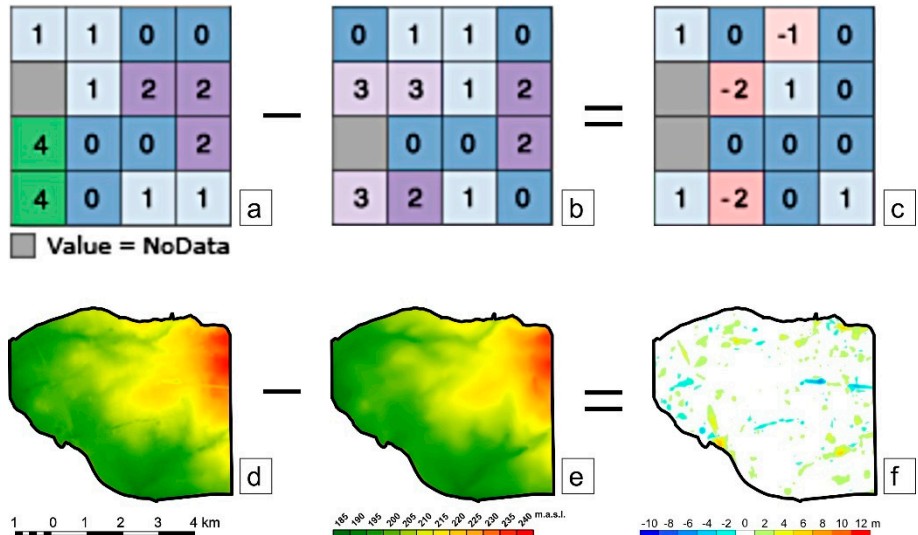

**Figure 8.** Diagram of how the raster image subtraction function (ESRI 2020) works, using data for Łódź as an example. (**a**) theoretical raster, (**b**) second theoretical raster, (**c**) subtraction result. (**d**) NMT 2017, (**e**) NMT 1917, (**f**) differential model. Source: Own study.

The model obtained in this way showed all possible changes in the land surface, both the actual ones and those resulting from measurement methods appropriate for data acquisition. In order to eliminate the latter, data of lower quality, i.e., those derived from the historical contour drawing, were chosen as a reference. These data had a vertical resolution of 1 m (contour cut 1 m), hence it was assumed that objects with a height of less than a m may not have been recorded on the historic map (a decision made by the surveyors mapping at the time). It was therefore decided to ignore changes in landforms smaller than 1 m in both positive and negative values in the differential model (Figure 9).

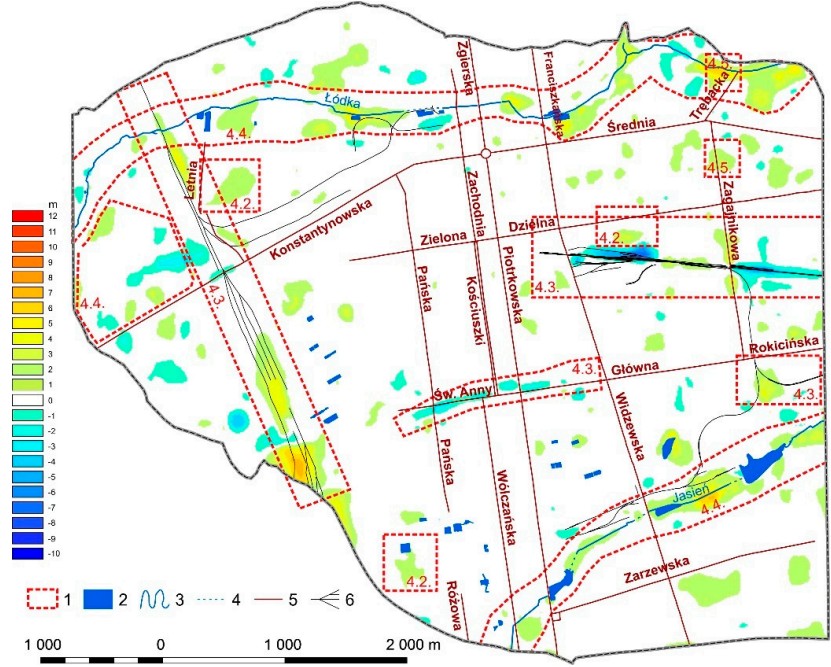

**Figure 9.** Changes in the relief of a part of Łódź against selected elements of the environment and the city's infrastructure as of 1917. 1—areas of detailed analysis, 2—waters, 3—surface watercourses, 4—underground watercourses, 5—main streets, 6—railway tracks (situation in 1917). Source: Own study.

## 4. Results

### 4.1. General Patterns of Relief Changes

On the differential maps presenting a fragment of Łódź in 1917 and 2017 against the background of selected contemporary elements of the city's environment and infrastructure (Figures 9 and 10), several large areas of superficial land elevations and several with zones of linear fairly deep depressions can be identified. Several characteristic cases were selected to analyse and explain in detail the changes in the relief of Łódź over a period of 100 years.

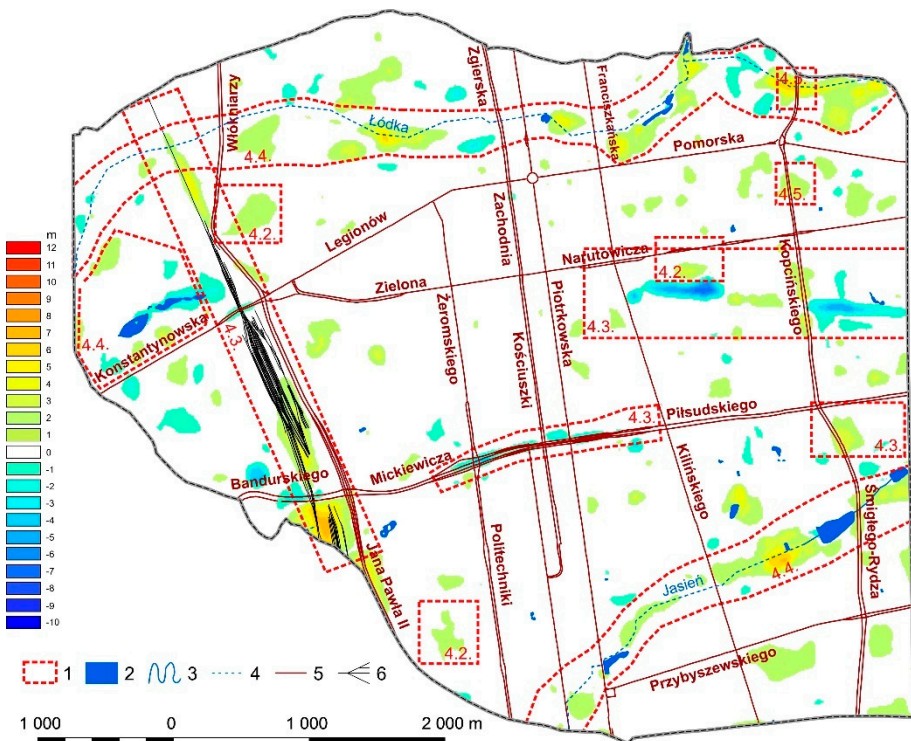

**Figure 10.** Changes in relief for a fragment of Łódź in 1917–2017 against contemporary elements of the city's environment and infrastructure. 1—areas of detailed analysis, 2—waters, 3—surface watercourses, 4—underground watercourses, 5—main streets, 6—railway tracks (situation in 2017). Source: Own study.

### 4.2. Elevations Resulting from the Construction or Demolition of Buildings

Most buildings in the study area of Łódź were constructed by the end of the 19th century [41]. Assuming that the morphometric picture is from a similar period (early 20th century) and that most of the development has survived to the present day, the relief of the land surface should not change. In fact, looking as a whole, the changes are definitely greater in the peripheral zones of the study area, but they also occur in the very centre (Figures 9 and 10). Interpretation of the observed changes in the relief of the city centre requires reference to its origins and observation of the processes of its layout and development.

Examples include the surroundings of today's Dąbrowskiego Square, the Grand Theatre and the Rector's Office of the University of Łódź (detailed study area 4.2, Figure 10). The authors set out to explain the origin of these changes (Figure 11). For this purpose, archival plans of the city and the application https://atlas.ltn.lodz.pl/aplikacja/, accessed on 14 June 2022, on which the analysed area is presented, were used.

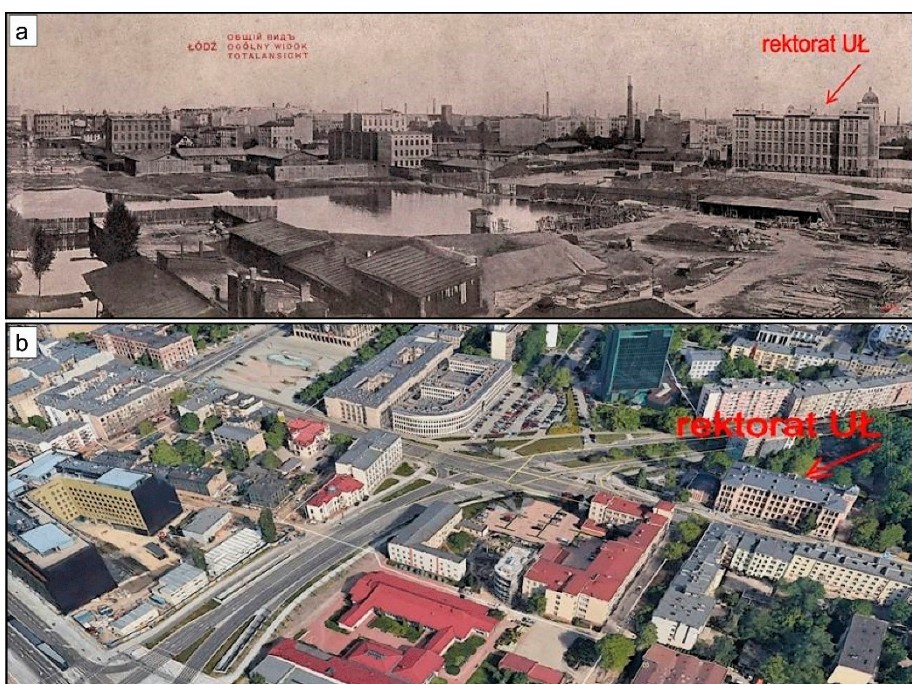

**Figure 11.** The pond at the mineral extraction site juxtaposed with today's image of the area. (The building that currently houses the Rector's Office of the University of Lodz is distinguished for the location). Source: (**a**) fotopolska.eu, 1910–1914, (**b**) Goggle Earth 2021.

The first city plans showing the area of today's Dąbrowski Square date back to the beginning of the 19th century, when Łódź was defined as a factory town and the first clothmakers' settlement, Nowe Miasto, was delineated. In 1823, a new street grid was drawn on a plan by Philippe de Viebig in 1823 (Figure 12a). The most important element was the octagonal market square, which was bisected by a road connecting Łódź with Łęczyca in the north and Piotrków Trybunalski in the south. An Evangelical church and a town hall were built on it. The analysed area—marked with a yellow polygon—was located in the south of the Nowe Miasto, between the newly delimited Cegielniana Street (currently Jaracza) and Dzielna Street (currently Narutowicza), laid out in 1830, where a municipal brickyard was established for construction purposes.

Until the end of the 19th century, the whole area of Nowa Dzielnica (New Quarter) was successively built up, but an analysis of the individual town plans shows that for many years, the area in question had a square with no buildings (Figure 12b). On Starzyński's plan (1897) (in Cyrillic characters), we can see a residential building in the eastern frontage, in the northern one a villa of the industrialist Jakub Icek Izaak Kestenberg. On the western side facing the square, there was an Old People's and Cripples' Home of the Lodz Christian Charitable Association (Figure 12c). The southern frontage of the market was still empty at the time. Until 1930, the square was used as a marketplace. It was the centre for the sale of the products of the nearby brickworks, and for this reason it was called Rynek Cegielniany (Brick Market) (Figure 13). This brickyard exploited mineral resources, contributing to the formation of the surrounding pits.

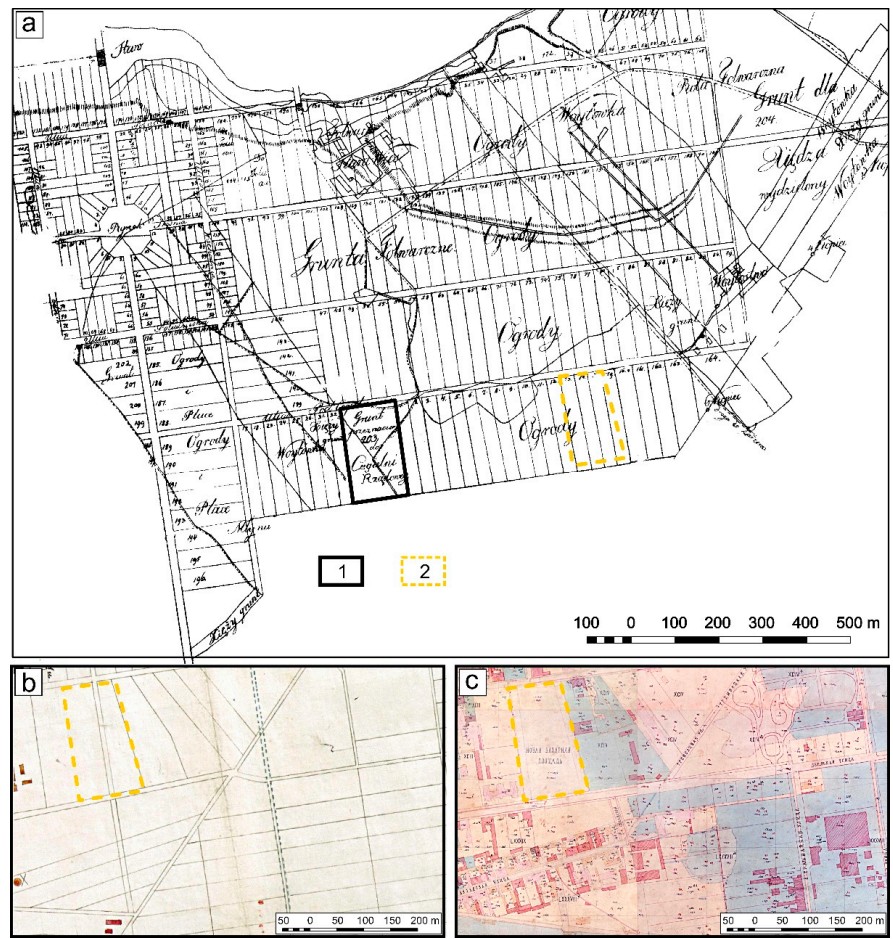

**Figure 12.** Changes in the relief of a fragment of the centre of Łódź. (**a**) plan of the clothmakers' settlement—Nowe Miasto by F. de Viebig from 1823, (**b**) vicinity of Dąbrowski Square on H. Majewski's plan (1889), (**c**) vicinity of Dąbrowski Square on Starzyński's plan (1897). 1—planned area of the Government Brickworks of 1823, 2—location of future Dąbrowski Square. Source: Own study based on https://atlas.ltn.lodz.pl/, accessed on 14 June 2022; http://www.historycznie.uni.lodz.pl/przestrzenny.htm, accessed on 14 June 2022.

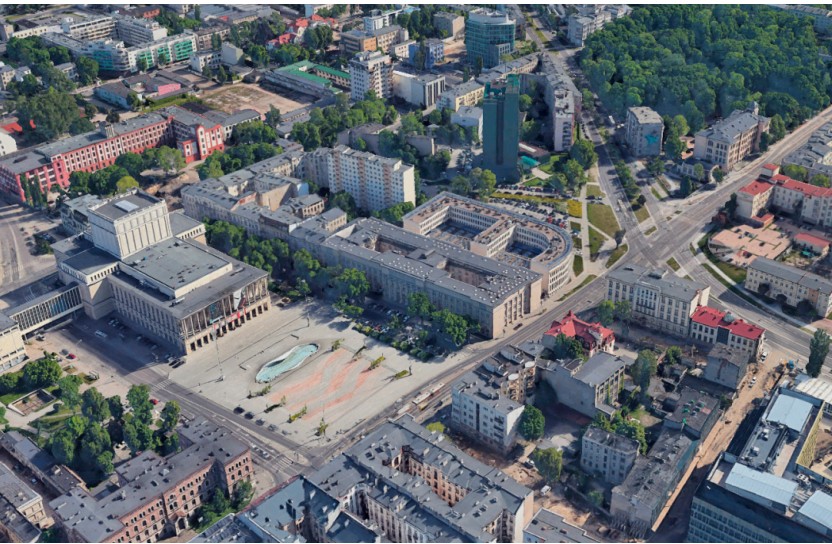

**Figure 13.** The contemporary landscape of Dąbrowskiego Square in Łódź. Source: https://earth.google.com/, accessed on 12 February 2022.

After nearly 100 years since the streets surrounding the square were laid out, it was decided to tidy it up and give it a new function. This related to Poland regaining its independence in 1918 and the creation of the new Łódzkie Voivodship. Łódź gained new administrative functions. At that time, the city authorities began efforts to change the appearance and function of the square. It was named after Jan Henryk Dąbrowski (a Polish general, among others, a participant in the Napoleonic Wars (1755–1818)). In the 1923 city plan, Dąbrowski Square is in the centre of the city but still does not play an important role.

In the eastern frontage of the square, the seat of the District Court (1927–1930) and residential buildings were built. From that moment on, its physiognomy changes. In the following years, the streets in the eastern and western frontages were laid out and corrected. In the years 1949–1967, an opera house was erected in its northern part, and in 2009, a 35-m fountain. Today, the square is surrounded by four streets and the following three frontages are built up: the court building, the opera house and townhouses (Figure 13).

Such significant changes in the development of the square over a period of more than 100 years may be responsible for changing its relief, i.e., for its elevation. It can be assumed that larger-scale development changes (other than paving and greenery changes) have been completed.

Other relatively large islands are visible in the study area, where there has been a clear elevation of the surface (Figure 14a). When cross-referencing with archival material, these changes coincide with the emergence of new housing estates or sites of transformation of already existing urban infrastructure. As an example, we can mention the Koziny housing estate (Figures 14a(I) and 15b). Originally, these were forested areas that were gradually transformed into orchards and gardens (Figure 15f). Over time, few buildings were built (Figure 15g). A breakthrough in the physiognomy of the area took place between 1960 and 1963 when a housing estate of multi-family buildings was built (Figure 15h). During their construction, there was a need for deep excavations for the foundations and basement level. The excavated earth layers were spread on the surface around the buildings, forming an overburden of varying thicknesses (Figure 14b).

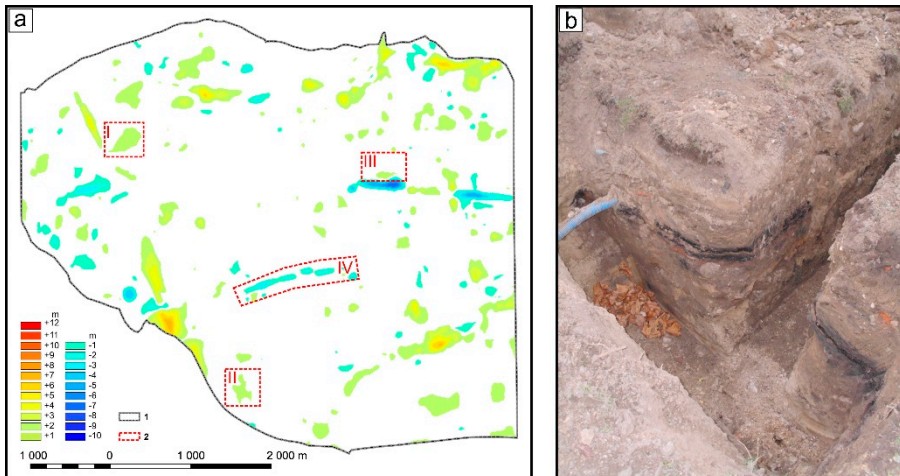

**Figure 14.** (**a**) Examples of relief changes due to new residential development and selected post-industrial buildings (1—boundaries covering the range of Jasiński's map; 2—areas of analysis); (**b**) Exposure showing land surfaces from before the construction of modern buildings; top surface of about 1 m (photo A. Szmidt 2021).

The raising of the area can also be linked to the demolition of existing buildings. As an example, consider the area in the south-western part of the site (Figure 14a(II)), where the textile factory of the French entrepreneurs Allart, Rousseau and Co. was located (Figure 15a,c–e). After its closure in 1989, the buildings were successively demolished for

new housing. The rubble has not been hauled away; it forms a small heap visible on the differential map.

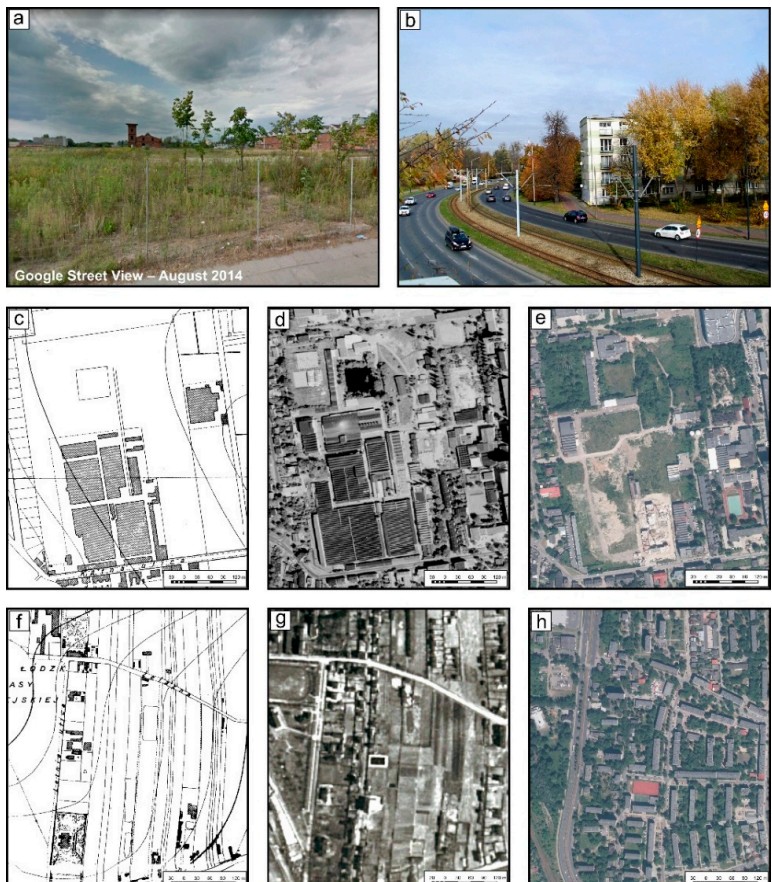

**Figure 15.** Examples of changes in the relief associated with the creation or liquidation of buildings for various purposes (photo. A. Szmidt 2021). (**a**) view of the area of the demolished Allart, Rousseau and Co. factory; (**b**) view of the housing estate in Koziny; (**c**) topographical situation in the factory area on Jasinski's map; (**d**) in the 1996 aerial photograph, (**e**) in the 2018 aerial photograph; (**f**) topographical situation of a fragment of the housing estate on Jasinski's map; (**g**) in the 1942 aerial photograph; (**h**) in the 2018 aerial photograph.

### 4.3. Changes in Relief Related to the Establishment and Operation of Railways and Roads

Significant changes in surface topography can be seen in the vicinity of the railway lines that ran through Łódź in the late 19th and early 20th centuries. Moreover, during the development of industrial Łódź several sidings were built from these lines, bringing the tracks to the more important textile factories (Figure 16a) [42]. Rail sidings no longer perform their functions. Some have been preserved as tourist attractions, while others have been dismantled. Visible trenches remain (Figure 16a(I,III)).

Over one hundred years, there have been major changes to the railway infrastructure that have altered the relief. Firstly, the areas around Łódź Kaliska and Łódź Fabryczna stations have been rebuilt. At the first one, a new traffic system was created. New railway tracks, roads and viaducts were built, which resulted in the appearance of a large embankment directly linked to the railway tracks in the northern part, while in the southern part, a vast area related to the modernisation of the station roads and crossings (Figure 16a(I),f,g).

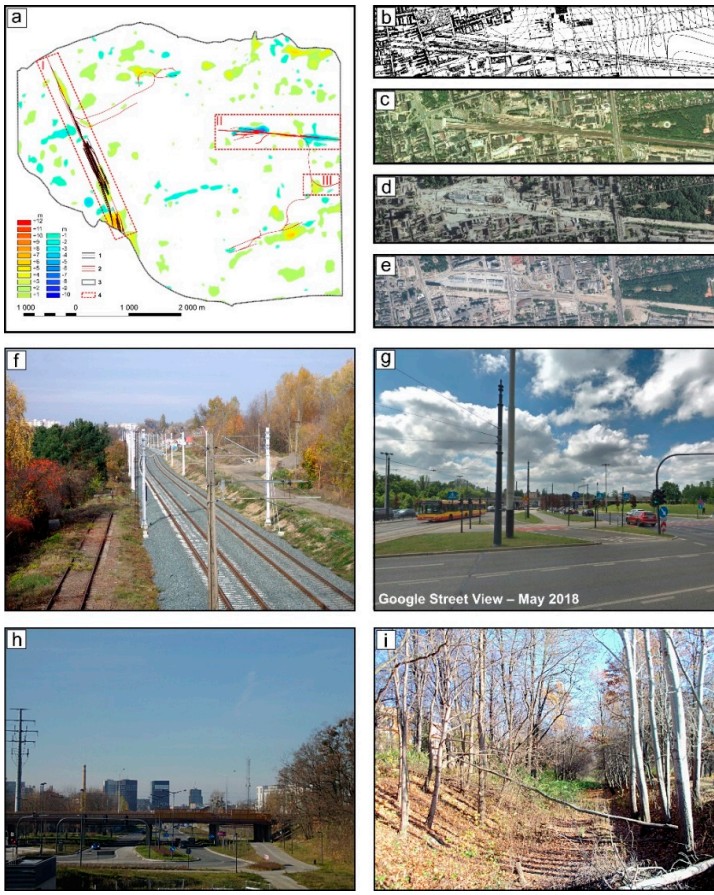

**Figure 16.** Changes in the relief of Łódź associated with the railway and road transport network (photo A. Szmidt 2021). Na mapie brakuje trasy WZ (**a**) location map (1—non-existing tracks; 2—tracks existing at the beginning of the 20th century; 3—limits of Jasiński's Map development; 4—areas of analysis); (**b**) fragment of the area of the present Łódź Fabryczna Railway Station on Jasiński's Map, (**c**) fragment of the area of the present Łódź Fabryczna Railway Station on the 1996 aerial photo; (**d**) fragment of the area of the present Łódź Fabryczna Railway Station on the 2015 aerial photo; (**e**) fragment of the area of the present Łódź Fabryczna Railway Station in an aerial photo taken in 2018; (**f**) view on the beginning of the railway embankment in the vicinity of Łódź Kaliska Railway Station; (**g**) view on the crossing near Łódź Kaliska Railway Station; (**h**) location of the former railway trench and tracks in the vicinity of Łódź Fabryczna Railway Station; (**i**) trench of the former railway line to K. Scheibler's factories.

Large linear depressions can be observed in the vicinity of Łódź Fabryczna station (Figure 16a(II),b–e). Between 2010 and 2016, the station was rebuilt, and the railway tracks were routed deep underground and led to the station buildings, which are also mostly underground. Most of the earth material was hauled away, while some was spread nearby. These depressions were caused by the modernisation of the railway line, while the elevations were the result of the modernisation of Kopcińskiego Street.

The long linear depression in the centre of the study area is the result of investment in pre-existing road infrastructure (Figure 16a, IV Św. Anny Street and Główna Street (currently Mickiewicza and Piłsudskiego Streets) were significantly widened in the 1960s (all buildings on the northern side of the street and some on the southern side were demolished). As a result of the investment, it had two carriageways with two lanes each and a separated tramway track in the middle Figure 14a(IV)). This section of the city's road network connects two very large housing estates and requires further investment to improve car and tram traffic. To this end, a large fragment of the road was rebuilt between 2017 and 2019. An interchange centre for tram lines running through the city centre was

built on Mickiewicza Avenue between Piotrkowska and Kościuszki Streets, and a tunnel for cars was constructed between Żeromskiego and Kilińskiego Streets.

### 4.4. Changes in Surface Topography in Areas of Former River Valleys

Łódź is not situated on a large river; however, 28 watercourses flow through it and are now mainly underneath it. It is estimated that the length of these watercourses in the area of modern Łódź is about 77 km. It is supposed that in the pre-industrial period, this length may have been up to three times longer [32,34]. Until regulation in the 1930s, the rivers flowed quite naturally (Figure 3c,d), providing water sources for industry and a transport network for discharged waste. The diversion of water to the canals required the deepening of river valleys and the routing of collectors. They were then backfilled and possibly overfilled [43]. With the development of the urban infrastructure, the area was subject to further levelling, which is shown by the elevation of the area on the resulting map. In addition, in the valley areas running through the city parks, ponds have been designed for the recreation of the residents. Their implementation caused both depression and elevation at the sites where sediments excavated for the construction of the reservoir were stored (Figure 17a).

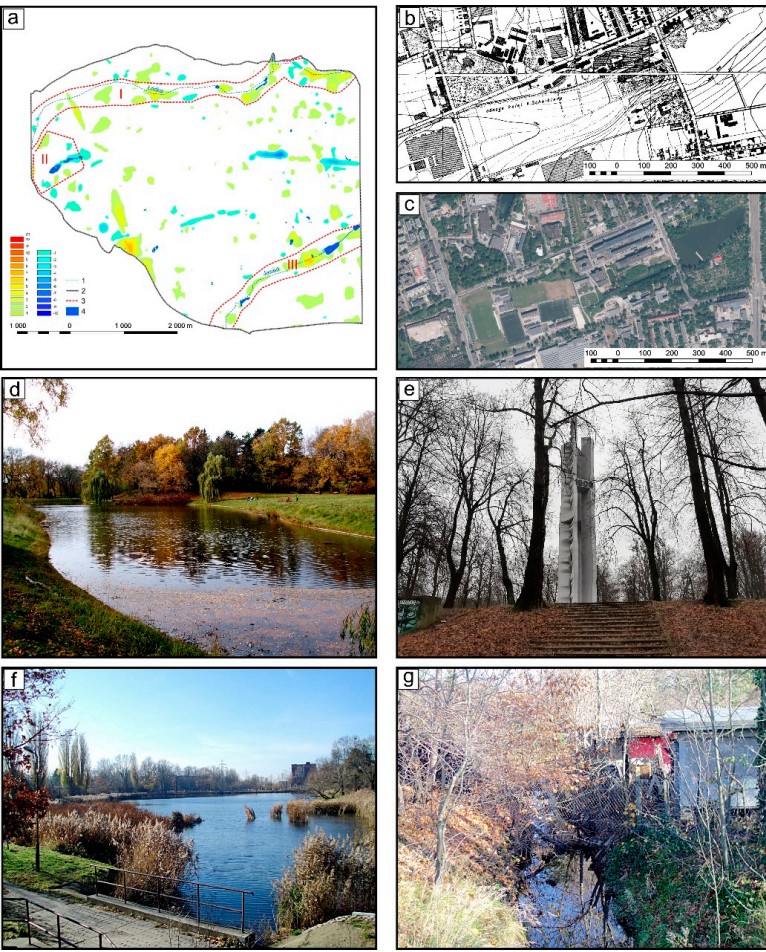

**Figure 17.** Changes in the relief of Łódź associated with the levelling of river valleys (photo A. Szmidt, I. Jażdżewska 2021). (**a**) location map (1—course of rivers at the beginning of the 20th century; 2—water bodies existing at the beginning of the 20th century; 3—borders of Jasiński's Map development; 4—areas of analysis; (**b**) topographic situation of a fragment of the Jasień valley on Jasiński's Map; (**c**) aerial photo of a fragment of the Jasień valley from 2018; (**d**) one of the ponds in Piłsudski Park; (**e**) An artificial hill with a monument in Piłsudski Park; (**f**) one of the ponds on the Jasień River; (**g**) fragment of the Jasień River canal.

Changes in surface topography are clearly visible in the areas of the former Łódka and Jasień river valleys. Łódka was one of the location factors for the agricultural town and its subsequent industrial development. The waters were used by the textile industry in the 19th century, and for this purpose damming was carried out in many places. Unfortunately, the continuous development of the city caused the water quality to be degraded very quickly and the riverbeds turned into sewage (Figure 18). Over the years, Łódka has been gradually channelled, its river valley filled up and built over. Now its bed has been placed in an underground tunnel and the surface of the land above it has been developed (Figures 6, 7 and 17a(I,III)).

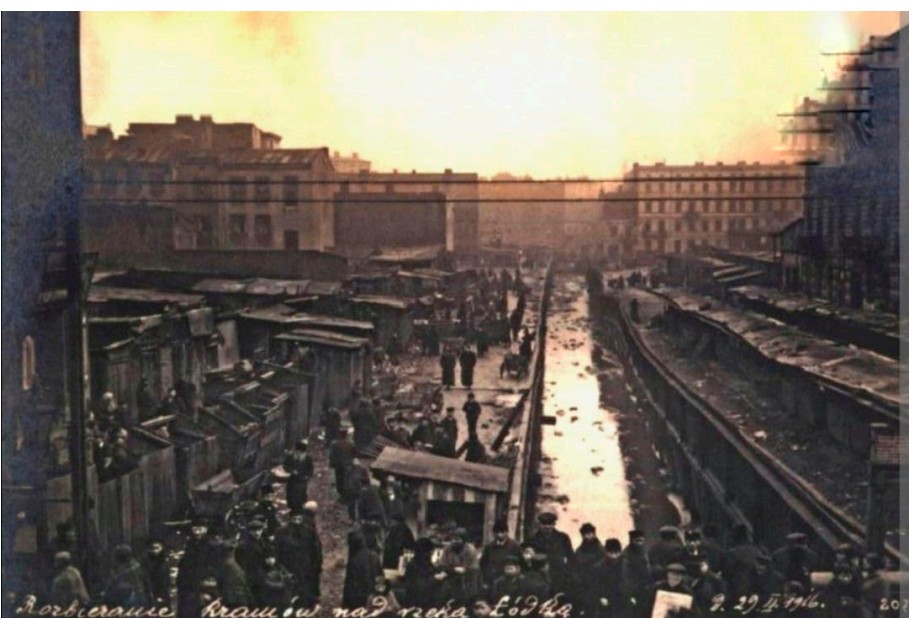

**Figure 18.** Łódka River—1916. Source: fotopolska.eu.

In the case of the Jasień River valley, similar processes took place as in the Łódka River valley the watercourse was channelled, some water reservoirs were left and some were liquidated (Figure 3c,d) (Figure 17a(III),b,c,f,g). On the differential model (Figure 17a(III)), an elevation of the surface in several sections of the former river valley formed after its canalisation and filling can be seen. At present, the valley is being successively developed (Figure 17b,c).

Depressions of the land related to the hydrographic network are visible in the northwestern part of the study area (Figure 17a(II)) in the area of the largest park in Łódź. The park design (Figure 19) used parts of the former urban forest and a small watercourse. A number of pathways have been laid out, new trees planted and a few ponds have been created on the basis of an existing watercourse (Figure 17d). To this end, the existing river valley has been widened and deepened as evidenced by depressions in the terrain, while earth from the resulting excavations has been spread or small hills have been built up, as seen in the resulting elevations (Figure 17a(II)). One of the elevations in the southern part of the park was formed after the Second World War. The Revolutionary Deed Monument, more than 30 m high, was placed there. To erect it, it was necessary to make a large trench and an earth embankment, on which the monument was placed (Figure 17e).

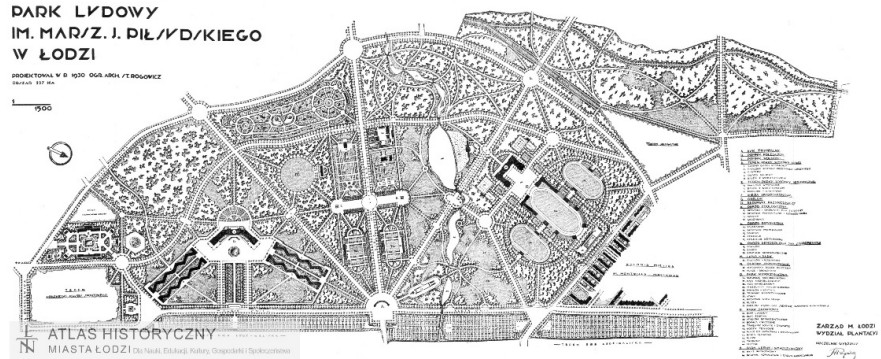

**Figure 19.** Design for Józef Piłsudski park called Park na Zdrowiu from 1930. Source: https://atlas.ltn.lodz.pl/imageatlas/07_09_A_1.jpg, accessed on 14 June 2022.

### 4.5. Changes in Surface Topography on Sites of Former Brickyards, Gravel Pits, Waste Dumps

In geological terms, the analysed area is built mainly of clays, sands and gravels [31]. Depending on the quality of the material, some sites were attractive local mineral extraction sites. It should be noted here, however, that during the period of the town's vigorous growth, smaller mining sites were often not formally recorded and are only visible on topographic maps in the form of small ditches [44]. Once the raw material had been extracted, various types of waste were deposited and later the area was covered with an insulating layer, usually in the form of earth and rubble. As land resources in the city dwindled, these areas were allocated to, for example, allotment gardens [45]. Two documented examples of this type of facility can be shown in the study area.

Elevated sites are located in the north-east of the study area (Figure 20a,b). The first represents the site of a former brickyard, which was filled in with waste and a layer of insulation, followed by the creation of allotments (Figure 20c). The second site is the remains of shelters built by the Germans during World War II. Later, road construction materials and waste were stored there (Figure 20d).

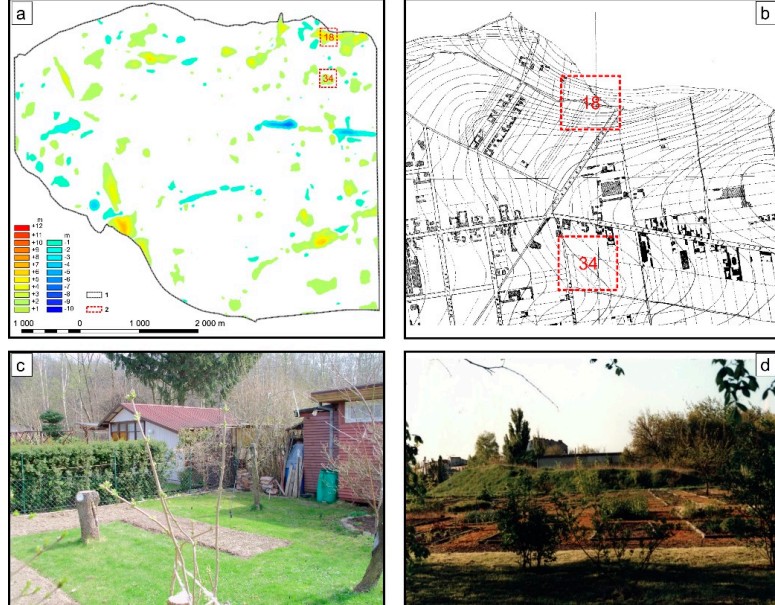

**Figure 20.** Changes in the relief of Łódź associated with mineral extraction and reclamation of excavations (photo Kuhn 1994). (**a**) location map (1—boundaries of Jasiński's Map; 2—areas of analysis); (**b**) topographic situation of the analysed area on Jasiński's Map; (**c**) site of former raw material extraction in Telefoniczna Street; (**d**) the site of former shelters in Kopcińskiego Street).

## 5. Discussion

The human impact on the environment is particularly evident in urban areas. One of its visible effects is changes in landforms. The authors decided to join the current of studying landform changes by combining the research experience of geomorphologists with that of socio-economic geographers. The main research problem was to investigate changes in relief in the area of the post-industrial city over more than one hundred years and to explain their causes. In order to achieve it, it was necessary to obtain archival and contemporary data and develop a methodology for comparing them.

The authors pointed out the problems that can occur when trying to compare archival 100-year-old cartographic resources with contemporary laser scanning data (LIDAR). The first difficulty lies in finding and acquiring maps containing elevation data of sufficient quality. It must be kept in mind that in the case of archival data, elevation data were obtained by direct measurements in the field, while contours were interpolated and plotted manually by cartographers (using traditional tools such as French curves). In addition, such measurements required a correspondingly long time, followed by recalculations, which limited their simultaneity. The situation is completely different in the case of data obtained from automatic measurements. They are acquired at a specific time interval, made with the same standardised equipment and then processed into a numerical terrain model using the same algorithms. In the case of digital models, to recreate this former concept of plotting the terrain surface, morphometric data can be subjected to smoothing and generalisation (e.g., with tools such as the following: Smoothing functions, spline interpolation or local polynomial). After processing the numerical terrain models for the ancient and modern areas, the calculated RMSE for the 1917 map was 0.43, which was not much worse than the RMSE calculated for the LIDAR data (0.36).

Only models with similar spatial resolution can provide reliable comparison material and form the basis for generating a differential map. One that minimises the risk of indicating changes that are the result of inaccuracies in the survey procedure rather than actual changes on the ground.

Therefore, the obtained numerical terrain models require verification with information from historical queries and field inventory.

The Numerical Terrain Model based on LIDAR data was too accurate relative to the model generated for the archival data. For this reason, laser scanning data (LIDAR) should be generalised to the quality of those available in archival studies. The differential maps obtained showed a picture of changes in the surface relief. The observation of the differential map allows for first reflections on the changes in relief. There were quite a few of these, and they could have been the result of measurement errors, so it was decided to disregard changes in relief smaller than 1 m in both positive and negative values in the differential model.

The results obtained suggest that adequate archival maps can be a very good source for such analyses. This study can be a source of encouragement and a starting point for future work using historical source data for spatial analyses, especially of relief changes in urban areas. Techniques for comparing surfaces of different ages, together with a historical analysis, can form the basis for wider application research related to the management of city space, including the renaturalization of river valleys or the prediction of geohazards associated with highly anthropogenically transformed areas.

As a result of using the differential map and historical queries, it is possible to infer changes in the relief of the urbanised area. The authors believe that a positive validation of the results can provide a recommendation for other researchers interested in studying relief changes in urban areas. The interdisciplinary character of the study should be emphasised. The interpretation of the results was carried out in a geographical-historical context.

This study fills gaps in the existing scientific literature on urban change over a century. The existing scientific literature has mostly focused on indicating and describing landform changes but has lacked geographic and historical interpretation [1–4,6,8]. This issue has been resolved by inviting a socio-economic geographer to join the research team.

It should be noted that the survey was conducted in only one city. The authors had good-quality archival data available. For studies of other urbanised areas, there may be difficulties in obtaining them. They are difficult to replace. The availability of modern LIDAR data is theoretically possible, but practically not every city has this type of data.

## 6. Conclusions

The paper presents a concept and a proposal for solving problems related to the analysis of changes occurring in urban space over 100 years, on the example of one of the Polish post-industrial cities. As a result of the research procedure (Figure 4), it was observed in Łódź as follows:

- In the city centre, the changes in surface relief were minimal. This may be interpreted by the fact that 100 years ago, the downtown area was already in the final stage of urbanisation. Only a few areas—Łódź Fabryczna railway station and Dąbrowskiego square—underwent visible transformations;
- Changes in the relief were the result of road and railway investments in the city (viaducts, tunnels, excavations, trenches);
- Changes in surface topography in the former river valley areas resulted from the redirection of water into canals and the creation of ponds for the recreation of the inhabitants in the valley areas;
- The changes in surface topography in the areas of former brickyards, gravel pits and waste dumps are a result of land levelling and their use, as for example, allotment gardens;
- Changes in the relief resulted from the construction or demolition of buildings (new housing developments, demolition of post-industrial buildings).

**Author Contributions:** Conceptualization, M.J. and A.S.; methodology, M.J. and A.S.; writing—original draft preparation, M.J., A.S. and I.J.; writing—review and editing, M.J., A.S. and I.J.; visualization, A.S. All authors have read and agreed to the published version of the manuscript.

**Funding:** This research received no external funding.

**Data Availability Statement:** Not applicable.

**Acknowledgments:** We would like to thank the anonymous reviewers for their valuable comments that had a positive impact on the final version of the article.

**Conflicts of Interest:** The authors declare no conflict of interest.

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
