# Peer review of "Changes in Land Relief in Urbanised Areas Using Laser Scanning and Archival Data on the Example of Łódź (Poland)"

_remotesensing, doi:10.3390/rs14132961_

Round 1

Reviewer 1 Report

The authors have addressed my previous comments only partially. The most important comment, related to the lack of the most important section, the Discussions, was not addressed. Instead of developing the missing section according to the suggested structure few paragraphs were added to the Results and, even worse, to the Conclusions. Apart from misplacing the newly added paragraphs, they are also insufficiently developed, and require a serious development for the publication of this manuscript. Due to the refuse of authors to address this crucial comment, the article is still lacking research depth, and has an unclear structure, impeding its understanding. I have reproduced my earlier comments, and require the authors to address them in full or provide a sound justification for not doing it.

"The authors lack the most important section of a research article, the Discussions, meant to emphasize the importance of research, justifying its publication. Normally, this section includes include (A) the significance of results - what do they say, in scientific terms; (B) the inner validation of results, against the study goals or hypotheses; (C) the external validation of results, against those of similar studies from other countries, identified in the literature; (D) the importance of the results, meaning their contribution (conceptual or methodological) to the theoretical advancement of the field; (E) a summary of the study limitations and directions for overcoming them in the future research. Out of these, only the significance of results is presented, but poorly, and there is a section labeled "Uncertainties" corresponding to the limitations. Only (A) is partially present. The "Discussions" should be developed to include the missing elements."

Author Response

Thank you very much for your more recent and previous comments, we have tried to amend the manuscript in such a way as to take into account all suggestions from 4 reviewers.
That is why we first focused on specific comments on showing the accuracy of the models or on the development of the literature.
In this step, we have rebuilt the last paragraph of Conclusions into the suggested 2 chapters. The manuscript currently has a Discussion and Conclusions chapter.
Thank you for your comments and we hope that our article will be approved after these changes.

Reviewer 2 Report

The paper has improved from the initial draft and can be published.

Author Response

Thank you very much for the present and previous comments, we tried to make changes to the manuscript in order to take into account all comments of 4 reviewers. 
Thank you again for your kind acceptance of our work.

Reviewer 3 Report

I do not have anything to add or suggest. The only thing that I should underline is the fact that the Authors gave really brief answer to my comments.

Author Response

Thank you very much for your presents and previous comments, we tried to make changes to the manuscript in order to take into account all comments of 4 reviewers. 
Thank you again for your kind acceptance of our work.

Round 2

Reviewer 1 Report

Although my comments were crystal-clear, the authors have addressed them superfluously. This can be seen in the sketchy response; although I provided a 5-points structure of the most important section, the discussions, this section remains unstructured, and their response does not indicate which new paragraphs are addressing each point. I guess that they miss the potential or willingness to improve their article. Although the article had some potential, it lacks the research depth needed for publication with "Remote Sensing", but the authors cannot or do not want to do more. Therefore, given their little improvements, I do not have further comments, since I am not confident about having them addressed.

This manuscript is a resubmission of an earlier submission. The following is a list of the peer review reports and author responses from that submission.

Round 1

Reviewer 1 Report

The article lacks research depth, and it cannot be accepted for publication in its current form. The literature review is too sketchy and descriptive to justify the publication in an international journal, and the article lacks a discussions section. Moreover, the organization of the manuscript, against the structure of a research article, makes its understanding impossible. This judgment is based on framing the article as a research one. If it were a case study, I would have recommended a major revision instead. Detailed comments are provided for each section apart.

The introduction is poorly written. The literature review is insufficient (expedited in 16 rows, lines 24-40, and 17 references), and not carried in a critical way, failing to identify the shortcomings of previous studies (ambiguities, controversies, misconceptions or lacks), justifying the need for research, and emphasizing the novel and original elements of the current research, placed in the context of the previous shortcomings. The introduction ends with a paragraph presenting the research goals, which are not ambitious enough, pertaining only to the case study, and not making a clear contribution to the theoretical advancement of the field, conceptual or methodological. It is followed in lines 43-51 by strange statements, pertaining to the methods and results, which do not fit the purpose of introduction: creating a research framework for the current research based on a critical literature review. These lines seem to me an unnecessary summary of the paper. Since the article has an abstract, and its role is to summarize the paper, a second summary is not necessary, and confuses the readers. Instead, the authors should state the novel and original elements of their research, and the research gap filled in by their study.

The next section deals with the study area; it is placed as a separate section, although the presentation of the case study, including the description of the study area, belongs to the methods.

The authors lack the most important section of a research article, the Discussions, meant to emphasize the importance of research, justifying its publication. Normally, this section includes include (A) the significance of results - what do they say, in scientific terms; (B) the inner validation of results, against the study goals or hypotheses; (C) the external validation of results, against those of similar studies from other countries, identified in the literature; (D) the importance of the results, meaning their contribution (conceptual or methodological) to the theoretical advancement of the field; (E) a summary of the study limitations and directions for overcoming them in the future research. Out of these, only the significance of results is presented, but poorly, and there is a section labeled "Uncertainties" corresponding to the limitations. Only (A) is partially present. The "Discussions" should be developed to include the missing elements.

The article has numerous inconsistencies. The first figure referred in the text is not 1, but 5 (line 48)! Although figures are labeled, as the journal requires, by Figure X, where X is their number, they are cited by Fig. X or Fig.X (inconsistent use of spaces against punctuation, and not matching in any case the journal style). The same inconsistency in using spaces in relationship to punctuation is present in numerous cases.

Reviewer 2 Report

In this paper the authors pointed out the problems that can occur when trying to compare archival 100-year-old cartographic resources with contemporary laser scanning data (LIDAR). The work clearly explains how to proceed to compare your past and current morphologies using the map algebra in ArcGIS and then verifying the derived map with respect to the transformations that the territory has had over time.
I would also like to suggest to the authors to compare what happened after the bombings of the Second World War and therefore after the reconstruction phase (see Affek et. al 2021 https://onlinelibrary.wiley.com/doi/full/10.1002/arp.1846), which certainly affected the change in the urban and natural layout of the study site.
I also suggest extending the general bibliography on methods used to compare multitemporal lidar data and historical cartography, see for example:

Lazzari M. 2020 - High-Resolution LiDAR-Derived DEMs in Hydrografic Network Extraction and Short-Time Landscape Changes. In: Gervasi O. et al. (eds) Computational Science and Its Applications – ICCSA 2020. Lecture Notes in Computer Science, vol 12250, pp 723-737. Springer, Cham. https://doi.org/10.1007/978-3-030-58802-1_52

The paper is publishable after these revisions.

Reviewer 3 Report

Dear Authors,

Here you can find my review related to the manuscript "Changes in land relief in urbanised areas using laser scanning and archival data on the example of Łódź (Poland)" by Jaskulski et al.

Overall, the effort of the authors is appreciable. However, there are many methodological gaps that need to be fullfilled prior to accepting the research in a Journal as Remote Sensing.

In detail, the paper addresses the study of morphlogical changes occured in a Poland area through comparing legacy and modern topographical data. Nowadays it is well known that the Digital Elevatuon Models (DEMs) error has to be accounted for during DEMs differencing operations. Many methods are now available in order to properly account for these uncertainties, and worth of noting are the researches of Prof. Lane and Prof. Wheaton. In the proposed research this foundamental step has not been carried out, as well as no mention on the alignment operation (i.e., in the correct registration of the two topographical datsets) is reported.

Furthermore, the Introduction section of the paper must be improved by a deeper background statement and by better highligthing the novel characteristic of the proposed research. In the same way, the methods section must be enhanced with a more detailed description of the undertaken procedures.

From my point of view, at the moment the research cannot be accepted for pubblication in a Journal as Remote Sensing. I would suggest to improve the manuscript with the suggested changes and then re-submit for a new peer-review process.

Best Regards.

Reviewer 4 Report

The authors are dealing with a really interesting topics. They have conducted qualitive work.

Nevertheless, they do not give any clue regarding the accuracy of their data. The matter of accuracy is one of the most important issues. They shlould claim in any case, the uncertinties of their analyses.

In addition, the authors should give some extra infromation regarding the external validation of their observations. E.g. did they test the height information against some height's benchmarks? Or against some GNSS measurements?

The auhtors ought to provide this kind of information.